# Attitude towards HPV Vaccination and the Intention to Get Vaccinated among Female University Students in Health Schools in Jordan

**DOI:** 10.3390/vaccines9121432

**Published:** 2021-12-03

**Authors:** Malik Sallam, Kholoud Al-Mahzoum, Huda Eid, Areej M. Assaf, Maram Abdaljaleel, Mousa Al-Abbadi, Azmi Mahafzah

**Affiliations:** 1Department of Pathology, Microbiology and Forensic Medicine, School of Medicine, The University of Jordan, Amman 11942, Jordan; m.abdaljaleel@ju.edu.jo (M.A.); ma.alabbadi@ju.edu.jo (M.A.-A.); mahafzaa@ju.edu.jo (A.M.); 2Department of Clinical Laboratories and Forensic Medicine, Jordan University Hospital, Amman 11942, Jordan; 3Department of Translational Medicine, Faculty of Medicine, Lund University, 22184 Malmö, Sweden; 4School of Medicine, The University of Jordan, Amman 11942, Jordan; Klo0180363@ju.edu.jo; 5School of Dentistry, The University of Jordan, Amman 11942, Jordan; hda0175066@ju.edu.jo; 6Department of Biopharmaceutics and Clinical Pharmacy, School of Pharmacy, The University of Jordan, Amman 11942, Jordan; areej_assaf@ju.edu.jo

**Keywords:** immunization, vaccine hesitancy, vaccine rejection, undergraduate students, sexually transmitted infection, knowledge, awareness, tumor

## Abstract

Cervical cancer is a leading cause of morbidity and mortality in women worldwide. The availability of prophylactic vaccines for high-risk types of human papillomavirus (HPV) infection represents an important advancement in the prevention of cervical cancer. In Jordan, the availability of the HPV vaccination is restricted to individuals who are willing to pay. The aim of the current study was to evaluate the willingness and attitude of female university students in health schools/faculties in Jordan to get HPV vaccination and their knowledge about the virus. A self-administered online questionnaire was distributed in October 2021, which comprised 27 items to evaluate HPV knowledge, history of HPV vaccination, intentions to get the HPV vaccine, and the reason(s) behind vaccine refusal for those who rejected vaccination. The study sample comprised 836 participants: medical students (39.7%), pharmacy students (26.0%), dental students (21.2%), and nursing students (13.2%). Only 524 participants had heard of HPV prior to the study (62.7%), of which 48.7% knew about the availability of HPV vaccines. The lowest level of HPV knowledge was observed among nursing students. Only 19/524 students reported a history of HPV vaccination (3.6%). The overall willingness to receive HPV vaccination if provided freely was 75.0%, while only 16.0% were willing to pay for the vaccine. The most common reason for HPV vaccine rejection was the perceived low risk to get HPV infection. Significantly higher intentions to get HPV vaccination were found among older participants and medical students. The embrace of vaccine conspiracy beliefs was associated with a significantly less willingness to get the HPV vaccination (*p* < 0.001). Dependence on the internet/social media as the source of HPV knowledge was associated with a significantly lower intention to get HPV vaccination (*p* = 0.002). The coverage of the HPV vaccination among female university students in health schools in Jordan appeared extremely low; however, three-fourths of the students who had heard of HPV were willing to receive the HPV vaccination if provided freely. Complacency appeared as a major factor for HPV vaccine rejection. Increasing the levels of knowledge and awareness of HPV infection and its association with cervical cancer through reliable sources is recommended. This can be helpful for the individual benefit of the students besides the potentially positive role they can play in community education. Countering vaccine conspiracy beliefs with proper education and awareness programs can be helpful to appraise the role of HPV vaccines in cancer prevention.

## 1. Introduction

Cervical cancer is considered a leading cause of morbidity and mortality among females worldwide [1]. Specifically, cervical cancer was reported as the fourth most commonly diagnosed cancer and the fourth leading cause of cancer-related mortality in women in 2020, with uneven distribution of its burden worldwide [2,3,4]. The most recent estimates on global cancer incidence and mortality pointed to 604,000 new cases of cervical cancer, with 342,000 mortalities from the disease as of 2020 [2].

Human papillomavirus (HPV) infection is among the most common sexually transmitted infections (STIs) worldwide [5,6]. It has been estimated that more than 80% of women and men acquire HPV by the age of 45 years [7]. The natural history of sexual HPV disease indicates a large fraction of subclinical infections, with the possibility of apparent infections manifested in anogenital warts (condylomas) in both sexes [8]. The persistence of high-risk HPV infection appears to play an essential role in cervical dysplasia with subsequent risk for cervical carcinoma [9]. Additionally, a number of contributing cofactors have been linked to an increased risk of cervical cancer, including a history of other STIs, cigarette smoking, and a high number of childbirths [10,11,12].

The vast majority of cervical cancer cases can be linked to infection by high-risk HPV types (e.g., HPV-16, HPV-18, HPV-31, or HPV-33), as evidenced through the detection of the virus DNA in almost all cervical cancers [9,13]. Additionally, the role of HPV as an oncovirus extends beyond its association with cervical cancer. This involves the recognition of HPV infection as a risk factor for the development of oral/oropharyngeal, vulvar, vaginal, anal, and penile cancers, besides a possible causal role in prostate cancer [14,15,16].

The huge burden of HPV-related cancers worldwide, with an absence of curative antiviral therapies necessitates a focus on the continuous development of therapeutic and preventive approaches to tackle such a public health issue [17]. Screening for premalignant changes involves the use of cytology-based Papanicolaou (Pap) smears, which were introduced in the 1950s [18]. Currently, a Pap smear is recommended every 3 years for women 21 to 65 years of age [19]. However, such an approach can be costly; therefore, HPV testing can provide an alternative approach in low-income settings [20,21]. In addition, cytology-based screening suffers from obvious caveats in relation to subjectivity, dependence on the skills of the examiner, and the need to maintain quality assurance measures besides the need for repeated testing [22].

Vaccination against high-risk HPV types represents the main preventive measure against HPV-related cancers [23]. Compared to screening for HPV-induced premalignant changes, vaccination provides the advantage of preventing infection by high-risk types rather than the mere detection of premalignant changes [24]. Subunit HPV vaccines containing the L1 protein of the virus, administered as three injections over six months, have shown remarkable efficacy and effectiveness profiles [25]. Three subunit HPV vaccine types have been approved so far; the bivalent Cervarix vaccine providing protection against HPV-16 and HPV-18, the quadrivalent Gardasil vaccine protecting from HPV-16 and HPV-18 besides the low-risk types HPV-6 and HPV-11, and the nonavalent Gardasil-9 vaccine (protecting against types: 6, 11, 16, 18, 31, 33, 45, 52, and 58) [26].

Vaccination against HPV is recommended in routine immunization programs starting at age 11 or 12 years, through to age 26 years [27]. The vaccination of older individuals might be considered; however, this approach is presumed to be less beneficial considering the higher possibility of previous exposure to HPV among this age group [28].

Therapeutic use of the HPV vaccination can also be considered, which entails targeting the clearance of infections that have already been established [29]. Nevertheless, more studies are needed to explore the potential value of this approach [30,31].

Scarce data on HPV infections and cervical cancer are available from Jordan. The latest report issued by the HPV Information Centre in October 2021 reported an annual cervical cancer diagnosis of 115 women with 71 mortalities from the disease in the country [32]. During 2000–2013, cervical cancer was ranked as the 11th most common cancer among women in Jordan and the 10th most common cancer among women aged 15–44 years [33].

Despite the previous evidence of the low prevalence of STIs in Jordan, updated studies are needed to assess the recent trends of such infections in the country [34]. The most recent study on HPV prevalence in cervical cancer in Jordan reported that 92% of the cases had evidence of the presence of HPV DNA, with HPV-16 as the most common type followed by HPV-18 [35]. The results of an earlier study on the prevalence and distribution of high-risk HPV in cervical carcinoma, low-grade, and high-grade squamous intraepithelial lesions in Jordanian women showed a similarly high prevalence of HPV [36].

A recent study that evaluated the prevalence of abnormal cervical smears in women at a specialized cancer center in the country (King Hussein Cancer Center) argued against the introduction of population-based HPV testing and vaccination [37]. This argument was based on the finding of a low prevalence of cervical cancers in the country, and the study suggested that a more widely available Pap smear testing might suffice [37]. Regarding the HPV prevalence in head and neck cancers, two recent studies in Jordan revealed that the isolation of HPV-16 was commonplace in these cancers in Jordan [38,39].

The previous survey studies addressing knowledge, attitude, and practices towards HPV-associated cancers have shown variable defects and gaps in Jordan, mainly regarding the awareness of HPV vaccine availability. Two recent studies among medical and dental students at the University of Jordan showed that a considerable number of students were not aware that HPV vaccines are available [40,41]. Additionally, a previous study conducted at the Jordan University of Science and Technology showed that only 45% of the study sample have ever heard about HPV vaccines [42]. Another recent study among medical students in Jordan showed that only 40.5% knew about the availability of HPV vaccines [43]. An earlier study showed that most of the gynecologists in Jordan have the intention to recommend HPV vaccines [44]. However, no previous studies have investigated the intention of female university students in health schools in Jordan to get HPV vaccination and its associated factors. The assessment of HPV knowledge and attitude towards HPV vaccination among this group can be beneficial to improve the educational programs in health schools/faculties in the country. Consequently, the improvements in HPV knowledge among university students can be helpful at the population level, considering the potential role of this group in community education.

Therefore, this study aimed to evaluate the overall intention of female university students in four health schools/faculties (medicine, dentistry, pharmacy, and nursing) to get vaccinated against HPV. In addition, the study sought to assess the overall level of HPV knowledge and its association with HPV vaccine acceptance. Finally, we aimed to assess the potential effect of embracing vaccine conspiracy beliefs on HPV vaccine acceptance.

## 2. Materials and Methods

### 2.1. Study Design

A self-administered questionnaire was developed based on a review of relevant previous studies addressing HPV knowledge and attitudes towards HPV vaccination, especially among female college students in various settings [45,46,47,48]. The survey was developed and presented to participants in Arabic and English languages concurrently.

The face validity of each item was checked by the first and the senior authors before the pilot test was conducted. The instrument was validated in a pilot test with six female respondents, which were not included in final analysis. There was a subsequent minor revision of the questionnaire following the pilot test.

The survey was distributed online using a snowball sampling approach starting with the contacts of the authors (four of whom are lecturers teaching courses for medical, dental, and pharmacy students, one senior medical student and one senior dental student). The survey link was also shared on different social media platforms (Facebook and Twitter) and the instant messaging application WhatsApp. Survey distribution took place in October 2021, with no incentives for participation.

The study was approved by the Department of Pathology, Microbiology and Forensic Medicine (meeting 02/2021/2022) and by the Scientific Research Committee at the School of Medicine/University of Jordan (reference number: 5083/2021/67). Informed consent was ensured by the presence of an introductory section of the survey used in this study, with a mandatory question asking for agreement from the respondent to participate in the study. All collected data were treated with full confidentiality.

The calculation of the minimum required sample size was conducted using the CheckMarket online sample size calculator [49]. Assuming a total of 20,000 female students at health schools in Jordan as of the academic year 2020/2021, with a 5% margin of error and a 95% confidence interval, the minimum sample size was 377 participants.

### 2.2. Survey Items

The survey comprised five sections as follows: the first introductory section included a short summary of the study aims with a mandatory item indicating the agreement to participate in this study. The second section comprised six items assessing demographic characteristics of the participants (age, ethnicity, nationality, school/faculty, university, and monthly income of the household). The third section comprised an introductory item to assess whether the participants had heard of HPV before the survey, with a request not to complete the survey if the answer was no. This was followed by eight items (correct/yes vs. incorrect/no) assessing HPV knowledge, with a request to answer based on the participant’s prior knowledge as follows: (1) HPV infections are rare; (2) HPV is sexually transmitted; (3) Genital warts are caused by HPV; (4) HPV can cause cervical cancer; (5) Men cannot get HPV; (6) Individuals can be infected by HPV for years without knowing; (7) HPV can be cured by antibiotics; and (8) Are you aware of the availability of HPV vaccines?

The fourth section comprised five questions as follows: (1) Have you received the HPV vaccination?; (2) Given that HPV vaccines are safe and effective to protect from infection by the most common HPV types associated with cervical cancer, are you willing to get the HPV vaccine is it is provided for free?; (3) If you answered the previous question with “no”, what is/are the main reason(s) for your refusal to get the HPV vaccine? (multiple selections were enabled for the following responses: (a) I consider these vaccines unsafe; (b) I consider these vaccines ineffective; (c) I do not consider myself at risk of HPV infection; (d) I do not consider HPV as a common infection in Jordan; (e) For me, it is inconvenient to be vaccinated; and (f) Visiting the doctor makes me feel uncomfortable, which keeps me from being vaccinated); (4) Are you willing to pay to get the HPV vaccination?; and (5) What was your main source of knowledge about HPV? (multiple selections were enabled for the following responses: (a) School courses; (b) University courses; (c) Television/radio/newspaper; (d) Family/friend communication; (e) Internet/social media outlets; and (f) Healthcare providers).

The final section comprised seven items of the previously validated vaccine conspiracy beliefs scale (VCBS) [46]. These items included: (1) Vaccine safety data is often fabricated; (2) Immunizing children is harmful, and this fact is covered up; (3) Pharmaceutical companies cover up the dangers of vaccines; (4) People are deceived regarding vaccine efficacy; (5) Vaccine efficacy data is often fabricated; (6) People are deceived regarding vaccine safety; and (7) The government is trying to cover up the link between vaccines and autism. Responses were allowed across a 7-point scale for each item (strongly disagree, disagree, somewhat disagree, neutral, somewhat agree, agree, or strongly agree).

### 2.3. Study Measures

The main outcome measure in this study was the willingness to receive HPV vaccines, with responses dichotomized as yes vs. no. The knowledge of HPV was assessed among the participants who had heard of HPV prior to this study, with the aforementioned eight items. Each correct/yes response was scored as one, while the incorrect/no responses were scored as zero, yielding a minimum score of zero and a maximum HPV knowledge score of eight. For the VCBS, a response with “strongly disagree” was given the minimum score of 1, and the maximum score of 7 was given to the “strongly agree” response with a subsequent, direct relationship between the VCBS and the general embrace of vaccine conspiracies. The internal consistency of the VCBS was shown by Cronbach’s alpha value of 0.901.

### 2.4. Statistical Analysis

Analysis was done through the International Business Machines (IBM) Statistical Package for the Social Sciences (SPSS) for Windows, Version 22.0. (Armonk, NY, USA: IBM Corp). Descriptive statistics (mean and standard deviation (SD)) were used to describe the scale variables (VCBS and HPV knowledge score). The association between categorical variables were evaluated using the chi-squared test (χ^2^). For continuous variables (HPV knowledge score and VCBS), analysis with categorical variables were assessed using the Mann–Whitney *U* test (M–W) and the Kruskal–Wallis (K–W) test. The association between vaccine conspiracy beliefs and the willingness to receive the HPV vaccination was assessed using multinomial logistic regression. The *p* value of 0.050 was assumed as the statistical significance cut-off.

## 3. Results

### 3.1. Characteristics of the Study Participants

In this study, a total of 836 participants formed the final study sample. General characteristics of the study sample stratified per school/faculty are provided in (Table 1). The participants were most commonly medical students (n = 332, 39.71%), followed by pharmacy students (n = 217, 25.96%), and dental students (n = 177, 21.17%), while the lowest number of participants was among nursing students (n = 110, 13.16%). All participants were Arabs in ethnicity, and the vast majority were studying at public universities (n = 800, 95.69%).

### 3.2. Prior Knowledge of HPV among the Study Participants

In the whole study sample, prior knowledge of HPV was reported by 524 participants (62.68%). Divided per school/faculty, the highest level of HPV prior knowledge was observed among medical students (75.00%), followed by dental students (63.84%), pharmacy students (57.60%), while the lowest level was seen among nursing students (33.64%, *p* < 0.001, χ^2^ test, Figure 1).

### 3.3. Overall Level of HPV Knowledge among the Students Who Have Heard of the Virus

In the whole study sample, the lowest level of HPV knowledge was observed for the item assessing HPV vaccination availability, with only 48.66% of the students being aware of the existence of HPV vaccines (Figure 2).

Stratified per school/faculty, significantly higher levels of knowledge were observed among the study participants for the following items: HPV infections are not rare (the highest percentage of correct responses was 72.57% among dental students, while the lowest percentage was 40.54% among nursing students, *p* < 0.001, Table 2); Awareness regarding the availability of HPV vaccines (the highest percentage was 61.85% among medical students, while the lowest percentage was 29.73% among nursing students, *p* < 0.001, Table 2); and individuals can be infected by HPV for years without knowing (the highest percentage was 93.57% among medical students, while the lowest percentage was 81.08% among nursing students, *p* = 0.027, Table 2).

Based on the 8-item HPV knowledge score, the study participants who had heard of HPV displayed a mean score of 6.30 (SD = 1.33). Knowledge was higher among Jordanian students (mean: 6.39 vs. 6.07, *p* = 0.019, M–W) and among medical students (mean: 6.61), compared to dental students (mean: 6.06), pharmacy students (mean: 6.04), and nursing students (mean: 5.78, *p* < 0.001, K–W). Additionally, HPV knowledge was higher among students with a higher household income (>1000 JOD) compared to those with a lower income (mean: 6.45 vs. 6.05, *p* < 0.001, M–W).

### 3.4. History of HPV Vaccination and the Intentions to Get Vaccinated

Among the study participants with previous knowledge of HPV prior to this survey (*n* = 524), only 19 students reported a previous history of HPV vaccination (3.63%). These participants were medical students (*n* = 10), dental students (*n* = 6), two pharmacy students, and a single nursing student. The highest percentage of previous history of HPV vaccination was observed among Palestinian students (5/18, 27.78%), followed by Kuwaiti students (5/92, 5.43%), and Jordanian students (9/375, 2.40%).

The intention to get HPV vaccination—if provided freely—was reported by a total of 393 students (75.00%). However, the willingness to pay for HPV vaccination was reported only among 84 students (16.03%).

The highest level of HPV vaccine acceptance was observed among medical students (*n* = 194, 77.91%), followed by pharmacy students (*n* = 94, 75.20%), dental students (*n* = 81, 71.68%), and nursing students (*n* = 24, 64.86%). However, this difference lacked a statistical significance (*p* = 0.282, χ^2^ test). The acceptance of the HPV vaccination was higher among students with a higher monthly household income; however, this difference lacked statistical significance (77.74% vs. 70.41%, *p* = 0.061, χ^2^ test).

The students that reported a willingness to receive the HPV vaccination displayed a significantly higher HPV knowledge score compared to those who reported unwillingness to get HPV vaccines (mean: 6.41 vs. 5.97, *p* = 0.004, M–W).

### 3.5. Factors Associated with Unwillingness to Get the HPV Vaccination

Among the study participants who expressed unwillingness to get the HPV vaccination if it is provided for free (*n* = 131), the most commonly cited reasons for vaccine rejection were those denoting complacency. Given that selection of multiple items was allowed, the most common reason for refusal of HPV vaccination was “I do not consider myself at risk of HPV infection” (90/131, 68.70%), followed by “I do not consider HPV as a common infection in Jordan” (43/131, 32.82%), and “For me, it is inconvenient to be vaccinated” (35/131, 26.72%, Figure 3).

### 3.6. Vaccine Conspiracy Beliefs Were Associated with a Lower Intent to Get HPV Vaccination

The mean VCBS for the participants who had heard of HPV prior to this study was 20.21 (SD = 8.40). A significantly higher VCBS was noticed among the students who rejected the HPV vaccination compared to those who were willing to receive it (mean: 24.02 vs. 18.93, *p* < 0.001, M–W).

The correlation between HPV vaccine acceptance with vaccine conspiracy beliefs and HPV knowledge was evaluated using multinomial logistic regression. The HPV knowledge score was divided into two categories based on the overall mean (>6.00 vs. ≤6.00), and the VCBS was also divided into two categories (≤20.00 vs. >20.00). The covariates used included: age (<20 years vs. ≥20 years); nationality (Jordanian vs. non-Jordanian); school/faculty (medicine vs. dental vs. pharmacy vs. nursing); and monthly income (≤1000 JOD vs. >1000 JOD). Higher VCBS was correlated with a statistically significant higher likelihood of HPV vaccine rejection (odds ratio: 2.33, 95% confidence interval: 1.52–3.56, *p* < 0.001), while the *p* value for HPV knowledge was 0.813.

### 3.7. Sources of Knowledge Regarding HPV among the Study Sample

The majority of students with prior knowledge on HPV pointed to university courses as their source of knowledge regarding the virus, with multiple selections being allowed (*n* = 271, 65.1%), followed by internet/social media outlets (*n* = 242, 46.2%), and healthcare providers (*n* = 109, 20.8%). The dependence on the following sources of knowledge regarding HPV were associated with more willingness to receive the HPV vaccination: university courses (271/341 (79.47%) vs. 122/183 (66.67%), *p* = 0.001, χ^2^ test); and healthcare providers (91/109 (83.49%) vs. 302/415 (72.77%), *p* = 0.021, χ^2^ test). On the contrary, dependence on the internet/social media outlets was associated with less willingness to get the HPV vaccination (166/242 (68.60%) vs. 227/282 (80.50%), *p* = 0.002, χ^2^ test, Figure 4).

## 4. Discussion

The noteworthy findings of this study that should be highlighted include: the extremely low level of HPV vaccine coverage among female students in the studied health schools/faculties in Jordan; the presence of approximately one-third of the participants with previous awareness of HPV who expressed an unwillingness to receive a free HPV vaccination; and the significant association between vaccine conspiracy beliefs with the lower intention to get HPV vaccination; the presence of major knowledge gaps regarding HPV in the study sample.

The low level of HPV knowledge was manifested by the presence of more than a third of the study participants who had not heard of the virus prior to the current survey study. This was particularly prominent among nursing students, who displayed an overall low level of HPV vaccine acceptance. In this study, only one-third of nursing students had heard of HPV. In contrast to this low level of knowledge, a recent study among nursing students in Turkey showed a high level of HPV knowledge; however, this was accompanied by a low level of HPV vaccine acceptance at a rate of only 14.4%, with a lack of confidence in efficacy as the most common cause for vaccine refusal [50]. Providing accurate information regarding the importance, safety, and effectiveness of the HPV vaccination in the prevention of cervical cancer can significantly improve the attitude towards these vaccines, as evidenced by a recent study among nursing students in Spain [51]. Herein, the level of HPV knowledge was higher among medical students compared to nursing students, and this was a recurring pattern observed in a recent study among students in India [52].

Recent studies in Jordan displayed the presence of noticeable gaps in HPV knowledge, particularly in relation to the presence of a large fraction of medical and dental students who lacked knowledge regarding the availability of HPV vaccines [40,41]. In a recent study among dental students at the University of Jordan, only 36.8% of the pre-clinical students and 44.0% of the clinical students knew that HPV vaccines exist [41]. Another recent study among medical students at the same university showed a greater level of knowledge regarding the awareness of HPV vaccine availability among clinical students (65.5%). This was consistent with our results that indicated a higher level of HPV knowledge among medical students compared to their dental, pharmacy, and nursing counterparts [40].

The major finding of this study was the extremely low HPV vaccine coverage (3.6%) among the study participants who heard of HPV prior to participation. Nevertheless, the willingness to receive HPV vaccines if provided freely was reported among three-fourths of respondents who knew about HPV. To put this result into a broader perspective, a recent study among young women in Saudi Arabia showed a slightly lower level of HPV vaccine acceptance (64.3%) [53]. Consistent with the results of this study, a very low level of HPV vaccine coverage (3.0%) was reported among females aged ≥18 years from four Arab countries, including Iraq, Jordan, Qatar, and the United Arab Emirates; however, the proportion of those who rejected the HPV vaccination was much lower at 4.2% (with 43.2% who showed willingness and 52.6% who were undecided) [54]. The percentage of those who received the HPV vaccination was slightly higher (16.5%) in an earlier study among female college students in Lebanon [55]. A comparable level of HPV vaccine acceptance was reported among female students in Nigeria (57.7%) [56]. The intention to get the HPV vaccination was lower (48.9%) among medical students in Saudi Arabia [57].

Cultural and religious factors can play a major role in attitudes towards vaccination [58]. In a majority of Arab countries, the discussion regarding STIs is considered a sensitive, often taboo topic [59]. This might result in low levels of knowledge and awareness regarding different aspects of these diseases. Additionally, this might negatively impact the recommendation of HPV vaccine uptake in the region [60].

On the other hand, the intentions to get HPV vaccination were extremely variable in countries with a majority of Muslims in South East Asian countries. A recent study by Khatiwada et al. showed that 95.8% of university students in Indonesia stated that they would receive the HPV vaccination [61]. Another study among health care students and professionals in Malaysia showed that only 15% of the participants would agree to get the HPV vaccination if it was provided for free, probably related to the inclusion of older participants [62].

A massive drop in the acceptance of HPV vaccination was noticed in the willingness to pay for the vaccine, which was also reported by a study among health students in Malaysia [63]. In Malaysia, HPV vaccination was included in the national immunization program for several years, which possibly contributed to higher knowledge and willingness to receive HPV vaccines, especially among younger generations [64]. Another recent study involving dental students in Saudi Arabia showed a slightly higher percentage of participants who previously received the HPV vaccination (12.0% among female dental students in the study) [65].

In relation to HPV knowledge, a striking finding was that a majority of the participants were unaware of the availability of HPV vaccines. A study among female college students in Lebanon showed a higher level of knowledge regarding the availability of such vaccines; however, it also showed poor to moderate knowledge in relation to the number of doses and price of these vaccines [55].

Among those who rejected the HPV vaccination in this study, complacency emerged as the most common contributing factor. This appears conceivable considering the previous evidence of a low incidence of cervical cancer and HPV infections in the country [33,34]. Complacency was also noticed in the study by Farsi et al., among dental students in Saudi Arabia, with 26% attributing HPV vaccine refusal for not being sexually active, and 25% stating that they did not need such a vaccine [65].

A reluctance to receive HPV vaccines has also been reported in various different studies. A study that involved female college students aged 18–26 in Michigan showed that 31.3% of the study sample did not intend to get the HPV vaccination, with concerns of safety and side effects as the most frequently cited reasons for vaccine refusal [66]. Low HPV vaccine coverage was also reported among college students in China, where only 9.5% of the females had received the vaccine [67]. Another recent study in 2019 among female college students in China showed a similarly low result (11.0%) [68]. The level of HPV awareness appeared low in Asian countries compared to European countries [69]. A substantial positive attitude towards the HPV vaccination was reported in a recent study among young adults in Italy [70].

One novel result of this study was the finding of a significant correlation between the embrace of general vaccine conspiracy beliefs and HPV vaccine rejection among female students in Arab countries. Previous studies have linked vaccine conspiracy beliefs with COVID-19 vaccine hesitancy, and this effect appears to extend beyond the recently approved vaccines targeting the control of the pandemic [71,72,73,74].

An interesting result of this study indicated that the dependence on healthcare providers and university courses were associated with significantly higher levels of vaccine acceptance; thus, highlighting the role of proper education and encouragement in cancer prevention. This pattern was also reported in a recent study from India, which praised the role of healthcare providers and classroom instructions in acceptance of the HPV vaccination [52]. Furthermore, two recent systematic reviews have shown that the strategies that could encourage the uptake of HPV vaccines included healthcare provider encouragement and narrative education regarding the beneficial role of such vaccines in cancer prevention [75,76].

Another important finding in this study was the correlation between the dependence on the internet/social media platforms with a lower intention to get HPV vaccines as opposed to the reliance on healthcare providers or university courses for HPV information. A previous study by Massey et al. reported on the possibility of the spread of misinformation/disinformation by anti-vaccination proponents using media channels such as Instagram and YouTube [77,78]. Thus, rigorous fact-checking is necessary as misinformation/disinformation can spread easily through such channels. This was evidenced by the association of reliance on social media with higher levels of embracing conspiracy beliefs and misinformation in the Middle East, which was noticed amid the ongoing COVID-19 pandemic [79,80].

Limitations of this study included the use of a convenience sampling approach, with the possibility of selection bias. Thus, the generalizability of the results should be approached with caution. In addition, survey studies entail the possibility of careless or multiple responses which cannot be ruled out. Sampling error that is related to the decision of the student to participate in the survey based on the previous knowledge of the survey subject should be considered as well. Thus, the genuine level of knowledge among students can be lower than the level reported in this study. The study participants (female students at health schools/faculties) are expected to have a higher level of knowledge compared to the general population, which was shown in recent studies that investigated COVID-19 knowledge [79,81]. Thus, the results of the current study might not be representative of the level of HPV knowledge and attitude towards the HPV vaccination among the general population in Jordan, and a study investigating such an aim is recommended.

## 5. Conclusions

Several gaps in HPV knowledge were identified in this study. The willingness of female students in health schools/faculties in Jordan to get HPV vaccination appeared acceptable if the vaccine was provided for free. The major reason behind HPV vaccine refusal was related to a low perception of being at risk of HPV infection. Rejection of the HPV vaccination was associated with the embrace of general vaccine conspiracy beliefs and reliance on the internet/social media as a source of knowledge regarding HPV. Thus, dismantling such beliefs and rigorous fact-checking such sources can be valuable to tackle hesitancy in HPV vaccine uptake. Special attention should be paid to incorporate refined educational courses for nursing students considering their low level of knowledge about the virus and its vaccination. The improvements in HPV knowledge among university students can also be beneficial at the population level, considering the potential role of such a group in community education.

## Figures and Tables

**Figure 1 vaccines-09-01432-f001:**
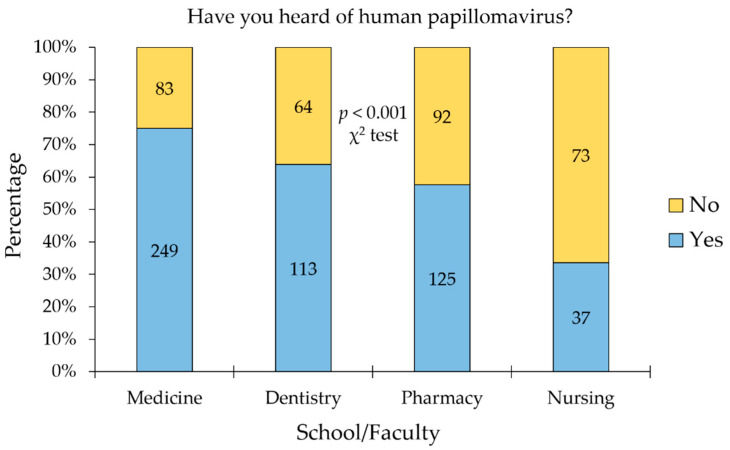
The prior knowledge of human papillomavirus among the study participants per school/faculty. The *p* value was calculated using the chi-squared (χ^2^) test.

**Figure 2 vaccines-09-01432-f002:**
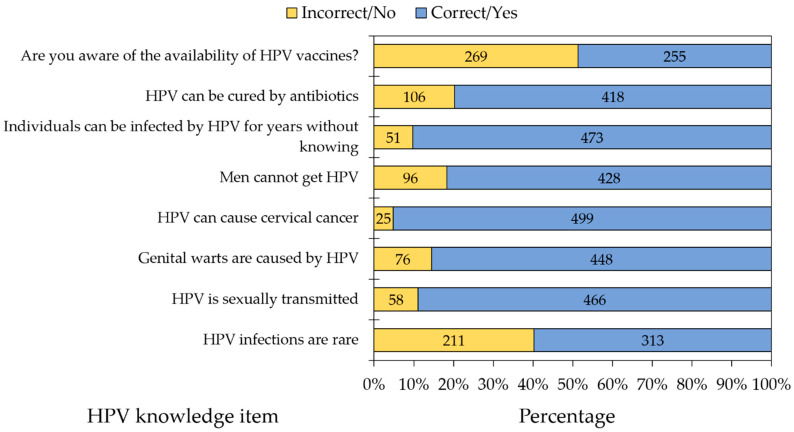
Overall knowledge regarding human papillomavirus among the study participants who have heard of HPV prior to the study (524/854), divided per knowledge item.

**Figure 3 vaccines-09-01432-f003:**
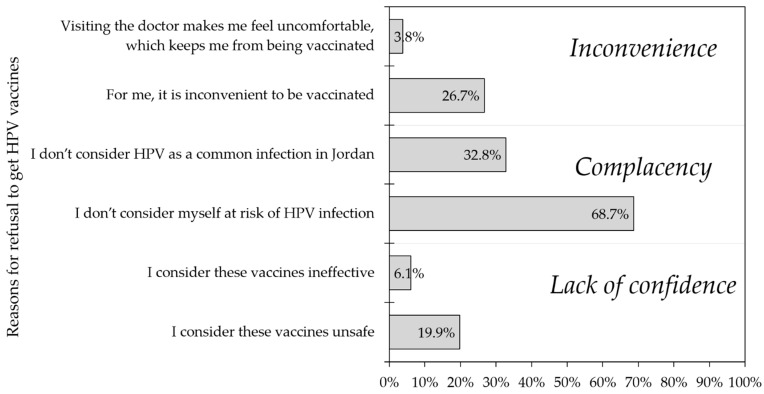
The reasons for refusal to get the human papillomavirus (HPV) vaccination.

**Figure 4 vaccines-09-01432-f004:**
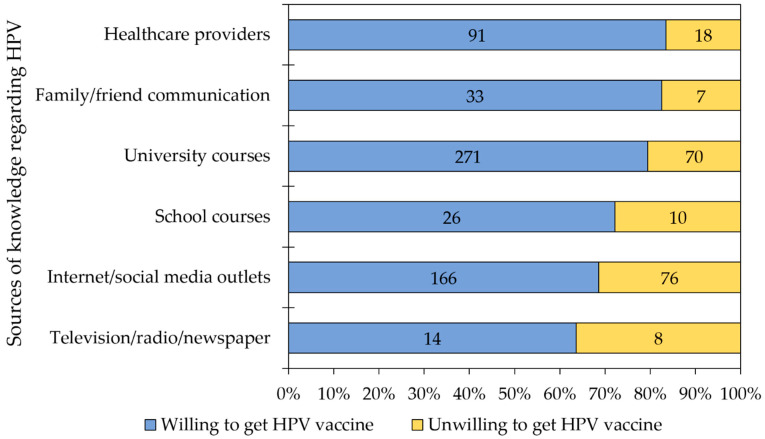
The sources of knowledge regarding the human papillomavirus virus among the study participants who had heard of the virus prior to the study. Multiple selections were allowed for each participant.

**Table 1 vaccines-09-01432-t001:** General features of the study participants divided by school/faculty.

Variable	Category	School/Faculty			
		Medicine	Dentistry	Pharmacy	Nursing
		n ^2^ (%)	n (%)	n (%)	n (%)
Age	Less than 20 years	124 (37.35)	76 (42.94)	25 (11.52)	42 (38.18)
	20–22 years	174 (52.41)	83 (46.89)	134 (61.75)	59 (53.64)
	More than 22 years	34 (10.24)	18 (10.17)	58 (26.73)	9 (8.18)
Nationality	Jordanian	197 (59.34)	106 (59.89)	177 (81.57)	107 (97.27)
	Kuwaiti	103 (31.02)	53 (29.94)	22 (10.14)	0
	Palestinian	10 (3.01)	11 (6.21)	4 (1.84)	3 (2.73)
	Iraqi	15 (4.52)	3 (1.69)	6 (2.76)	0
	Syrian	2 (0.60)	2 (1.13)	5 (2.30)	0
	Other (Arab)	5 (1.51)	2 (1.13)	3 (1.38)	0
University	Public	332 (100.00)	177 (100.00)	182 (83.87)	109 (99.09)
	Private	0	0	35 (16.13)	1 (0.91)
Monthly income of household	Less than or equal to 1000 JOD ^1^	87 (26.20)	66 (37.29)	129 (59.45)	85 (77.27)
	More than 1000 JOD	245 (73.80)	111 (62.71)	88 (40.55)	25 (22.73)

^1^ JOD: Jordanian dinar; ^2^ n: Number.

**Table 2 vaccines-09-01432-t002:** Human papillomavirus (HPV) level of knowledge stratified based participant affiliation to different schools/faculties.

HPV Knowledge Item	Response	School/Faculty				*p* Value ^2^
		Medicine	Dentistry	Pharmacy	Nursing	
		n ^1^ (%)	n (%)	n (%)	n (%)	
HPV infections are rare *	Correct	157 (63.05)	82 (72.57)	59 (47.20)	15 (40.54)	<0.001
	Incorrect	92 (36.95)	31 (27.43)	66 (52.80)	22 (59.46)	
HPV is sexually transmitted	Correct	228 (91.57)	98 (86.73)	106 (84.80)	34 (91.89)	0.186
	Incorrect	21 (8.43)	15 (13.27)	19 (15.20)	3 (8.11)	
Genital warts are caused by HPV	Correct	220 (88.35)	92 (81.42)	102 (81.60)	34 (91.89)	0.116
	Incorrect	29 (11.65)	21 (18.58)	23 (18.40)	3 (8.11)	
HPV can cause cervical cancer	Correct	238 (95.58)	106 (93.81)	120 (96.00)	35 (94.59)	0.857
	Incorrect	11 (4.42)	7 (6.19)	5 (4.00)	2 (5.41)	
Men cannot get HPV *	Correct	209 (83.94)	86 (76.11)	104 (83.20)	29 (78.38)	0.301
	Incorrect	40 (16.06)	27 (23.89)	21 (16.80)	8 (21.62)	
Individuals can be infected by HPV for years without knowing	Correct	233 (93.57)	97 (85.84)	113 (90.40)	30 (81.08)	0.027
	Incorrect	16 (6.43)	16 (14.16)	12 (9.60)	7 (18.92)	
HPV can be cured by antibiotics *	Correct	207 (83.13)	83 (73.45)	102 (81.60)	26 (70.27)	0.076
	Incorrect	42 (16.87)	30 (26.55)	23 (18.40)	11 (29.73)	
Are you aware of the availability of HPV vaccines?	Yes	154 (61.85)	41 (36.28)	49 (39.20)	11 (29.73)	<0.001
	No	95 (38.15)	72 (63.72)	76 (60.80)	26 (70.27)	

^1^ n: Number; ^2^ *p* value: Calculated using the chi-squared test; the incorrect items are marked with an asterisk. Analysis included the study participants who have heard of HPV prior to the study (524/854).

## Data Availability

The data presented in this study are available on request from the corresponding author (M.S.).

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
