# Peer review of "Attitude towards HPV Vaccination and the Intention to Get Vaccinated among Female University Students in Health Schools in Jordan"

_vaccines, 2021, doi:10.3390/vaccines9121432_

Round 1

Reviewer 1 Report

The study aims to diagnose the attitude towards HPV vaccination among female university students in health  schools in Jordan. This information is potentially useful to design educational programs to promote vaccination.

In general, the authors carry out an adequate study design, however, I consider that the selected sample - university students belonging to the health sector, who must have some level of information higher than the general population - may not be representative of the general population , limiting the scope of the real situation in the country. The authors somehow identified that weakness in their research.

At least with the study, the existing difficulties in this specific sector were identified, which are those that should develop the work of education for health in the population. This last idea should be highlighted both in the introduction and in the conclusions.

Author Response

Reviewer #1 comment

  1. The study aims to diagnose the attitude towards HPV vaccination among female university students in health schools in Jordan. This information is potentially useful to design educational programs to promote vaccination.

Response: We are deeply grateful for the reviewer positive view on this manuscript.

In general, the authors carry out an adequate study design, however, I consider that the selected sample - university students belonging to the health sector, who must have some level of information higher than the general population - may not be representative of the general population , limiting the scope of the real situation in the country. The authors somehow identified that weakness in their research.

Response: We are thankful for the meticulous comment of the reviewer. We agree with the point that university students, particularly in health schools/faculties are expected to have a higher level of information regarding the topic of active immunization against HPV for the protection of cervical cancer. Nevertheless, the selection of the students affiliated to heath schools can be very valuable as mentioned in the abstract “This can be helpful for individual benefit of the students besides the potential positive role they can play in community education.”

However, we agree with the reviewer’s comment; thus, we added the following sentence to the limitations section (Pages 12 & 13, lines 443-448): “The study participants (female students at health schools/faculties) are expected to have a higher level of knowledge compared to the general population, which was shown in recent studies that investigated COVID-19 knowledge [79,81]. Thus, the results of the current study might not be representative of the level of HPV knowledge and attitude towards HPV vaccination among the general population in Jordan, and a study investigating such an aim is recommended.”

  1. At least with the study, the existing difficulties in this specific sector were identified, which are those that should develop the work of education for health in the population. This last idea should be highlighted both in the introduction and in the conclusions.

Response: We are thankful for this important note by the reviewer, and accordingly we added the following sentences to the introduction section and the conclusions section:

Page 3, lines 128-132: “Assessment of HPV knowledge and attitude towards HPV vaccination among this group can be beneficial to improve the educational programs in health schools/faculties in the country. Consequently, the improvements in HPV knowledge among university students can be helpful at the population level, considering the potential role of this group in community education.”

Page 12, lines 459-461: “The improvements in HPV knowledge among the university students can be beneficial at the population level as well, considering the potential role of such a group in community education.”

Reviewer 2 Report

An interesting manuscript highlighting the knowledge of young female health sciences students, located in Jordan, however coming from numerous Arab countries.

The study is conducted on the basis of a well-designed questionnaire, the sample size is adequate and the results are in line with the questionnaire. The various weaknesses are already identified and highlighted in the discussion, thus the readers are aware of any limitations.

My sole comment is that in the introduction would be interesting to present information related to the cervical cancer burden in Jordan (for example prevalence, if available from National sources).

Otherwise, in my opinion, the manuscript is well written and presented, is of interest to the readers of Vaccines, and highlights reasons for non-vaccination especially in the COVID-19 period.

Author Response

Reviewer #2 comments

  1. An interesting manuscript highlighting the knowledge of young female health sciences students, located in Jordan, however coming from numerous Arab countries.

The study is conducted on the basis of a well-designed questionnaire, the sample size is adequate and the results are in line with the questionnaire. The various weaknesses are already identified and highlighted in the discussion, thus the readers are aware of any limitations.

Response: We are deeply thankful for the positive critical appraisal of our manuscript.

  1. My sole comment is that in the introduction would be interesting to present information related to the cervical cancer burden in Jordan (for example prevalence, if available from National sources).

Response: We would like to thank the reviewer for the insightful comment; however, we conducted a thorough literature review regarding the burden of cervical cancer in the country using the most up-to-date available date, as presented in the manuscript (lines 98-102). The scarcity of data precluded further elaboration on the important point raised by the reviewer.

  1. Otherwise, in my opinion, the manuscript is well written and presented, is of interest to the readers of Vaccines, and highlights reasons for non-vaccination especially in the COVID-19 period.

Response: Again, we would like to thank the reviewer for the positive critical appraisal of the manuscript. Thank you.